# Is 20 Hz Whole-Body Vibration Training Better for Older Individuals than 40 Hz?

**DOI:** 10.3390/ijerph182211942

**Published:** 2021-11-13

**Authors:** Shiuan-Yu Tseng, Chung-Po Ko, Chin-Yen Tseng, Wei-Ching Huang, Chung-Liang Lai, Chun-Hou Wang

**Affiliations:** 1Graduate Institute of Service Industries and Management, Minghsin University of Science and Technology, Hsinchu 30401, Taiwan; sytseng@must.edu.tw; 2Department of Neurosurgery, Tungs’ Taichung MetroHarbor Hospital, Taichung 43503, Taiwan; cbko1218@gmail.com; 3Department of Physical Therapy, Upright Come Scoliosis Clinic, Hsinchu 30286, Taiwan; uprightcome@gmail.com; 4Department of Physical Medicine and Rehabilitation, Taichung Hospital, Ministry of Health and Welfare, Taichung 40343, Taiwan; fredaex@gmail.com; 5Department of Occupational Therapy, College of Medical and Health Science, Asia University, Taichung 41354, Taiwan; laipeter57@yahoo.com.tw; 6Department of Physical Medicine and Rehabilitation, Puzi Hospital, Ministry of Health and Welfare, Chiayi County 61347, Taiwan; 7Department of Physical Therapy, Chung Shan Medical University, Taichung 40201, Taiwan; 8Physical Therapy Room, Chung Shan Medical University Hospital, Taichung 40201, Taiwan

**Keywords:** exercise, age groups, electromyography, rectus femoris

## Abstract

In recent years, whole-body vibration (WBV) training has been used as a training method in health promotion. This study attempted to use WBV at three different frequencies (20, 30, and 40 Hz) with subjects from different age groups to analyze the activation of the rectus femoris muscle. The subjects included 47 females and 51 males with an average age of 45.1 ± 15.2 years. Results indicated significant differences in subjects from different age groups at 20 Hz WBV. Muscle contraction was greater in the subjects who were older (F_(4,93)_ = 82.448, *p* < 0.001). However, at 30 Hz WBV, the difference was not significant (F_(4,93)_ = 2.373, *p* = 0.058). At 40 Hz WBV, muscle contraction was less in the older subjects than in the younger subjects (F_(4,93)_ = 18.025, *p* < 0.001). The spectrum analysis also indicated that at 40 Hz there was less muscle activity during WBV in the older subjects than in the younger ones. Therefore, age was found to have a significant effect on muscle activation during WBV at different frequencies. If the training is offered to elderly subjects, their neuromuscular responses to 20 Hz WBV will be more suitable than to 40 Hz WBV.

## 1. Introduction

Whole-body vibration (WBV) training is a neuromuscular training method. In recent years it has become popular in health promotion centers and gyms as an alternative training method or as a supplement to traditional training and treatment [1]. In addition, WBV has been comprehensively applied to various populations, such as young adults, middle-aged adults, and older adults [2,3], to achieve positive effects on neuromuscular performance, such as increased muscle strength [4], increased muscle endurance [5], and increased electromyography (EMG) activities [6].

It has been verified that the WBV training process can increase the neuromuscular activity recorded by surface EMG [7]. Although no consensus has been reached on the mechanism of the effect of WBV training on the neuromuscular system, many studies suggest that WBV training triggers the mechanism of the stretch reflex. This causes an excitatory response in the muscle spindles to induce a large amount of activity in the motor units [8]. Many studies have confirmed that the vibration frequency parameters of a body structure subjected to WBV training have an important effect on neurophysiological responses [9]. In an analysis of the effects of WBV characteristics on neuromuscular performance, Luo et al. (2005) suggested that the most effective frequency for generating greater muscle activity is between 30 and 50 Hz [10]. However, some studies have indicated that the maximum acceleration of the knee and hip occurs at approximately 15 Hz, and then decreases as the vibration frequency is increased [11].

Although WBV training has been comprehensively used in sports training and rehabilitation, no consensus has been reached on how WBV training induces neurophysiological responses in skeletal muscle [12]. Some studies have suggested that increased muscle activity during WBV training is associated with higher vibration frequencies and a greater vibration machine amplitude [13]. On the contrary, a study by Cardinale and Lim showed that compared with 40 and 50 Hz, a WBV frequency of 30 Hz provided the largest EMG activities of the vastus lateralis muscle [6].

A previous study showed that the median frequency (MF) and mean power frequency (MPF) slopes in the electromyography activities of elderly males during muscle endurance tests consisting of back stretching were significantly lower than those of young males [14]. This suggests that the corresponding muscle fibers in older adults are different from those in young people. Therefore, it would indeed be worthwhile to investigate whether WBV training may cause different neuromuscular responses in young people and in older adults. Previous studies have seldom investigated or tested the neuromuscular signal performances of all age groups during WBV training at different frequencies. Therefore, this study recorded the electromyography activities on the rectus femoris muscle of different age groups using WBV at 20, 30, and 40 Hz. These frequencies are often selected for WBV training and were analyzed to determine whether any responses were different.

## 2. Materials and Methods

### 2.1. Subjects

This study was one part of a research project titled “Effects of Vibration Training on Physiology in the Human Body”. The research project was conducted according to the Declaration of Helsinki and was approved by the hospital’s Institutional Review Board for research involving human subjects (Taichung Hospital, Ministry of Health and Welfare, Taiwan, I980021). All of the participants were fully informed of the study’s content policy before they participated, and all signed informed consent forms. The registered clinical trial number is ChiCTR-ICR-15006239. The subjects enrolled in the study were hospital volunteers and community members. All the subjects were healthy. None of the subjects had a history of neuromuscular disease, and none of the subjects were diagnosed with illness at the time of the investigation. The subjects were divided into groups according to their age: 20–29 years old; 30–39 years old; 40–49 years old; 50–59 years old; and 60–69 years old. There were a total of 5 groups.

### 2.2. WBV Training

The instrument used in this study was a whole-body vibration training machine (Commercial Grade Vibration Machine LV-1000; X-Trend, Taiwan). The vibration patterns had a peak-to-peak amplitude of 4 mm with a synchronous vibration platform. The frequencies were 20 Hz, 30 Hz, and 40 Hz. Participants received training for five minutes each time. Each frequency of whole-body vibration was randomly executed on different days. The subjects stood barefoot and without any support on the vibration platform with the knees in slight flexion position (10–15°).

### 2.3. Measurement of the EMG Signals of Rectus Femoris Muscle

The surface EMG signals were recorded from the rectus femoris muscle of the dominant leg to represent the knee extension muscle by a Noraxon EMG system (Noraxon Telemyo 2400T G2, Scottsdale, AZ, USA). Bipolar surface electrodes (Ag/AgCl) were applied over the belly muscle (interelectrode distance 25 mm) in accordance with SENIAM recommendations [15]. The pre-amplified EMG signals were amplified (×1000), band-pass filtered at 10–500 Hz ± 2% cut-off (Butterworth/Bessels), and sampled at 1500 Hz. MyoResearch software (Noraxon, Scottsdale, AZ, USA) was used to collect and store the data for analysis. 

#### 2.3.1. Measurement of the EMG Signals of Maximal Voluntary Isometric Contraction (MVIC)

A Biodex dynamometer (Biodex System 4 Pro, Biodex Medical Systems Inc., Shirley, NY, USA) was used to provide appropriate resistance against the MVIC; subjects were in a sitting position with knee flexion of 30 degrees [16]. The Noraxon EMG system was used to record the muscle activity of the rectus femoris muscle during MVIC [17]. Each subject performed the MVIC three times consecutively for five seconds each time, with three-minute rests between trials. The average value was recorded as EMG_MVIC_.

#### 2.3.2. Measurement of the Muscle Activity during Static Standing and WBV

The EMG signals were recorded once the subjects had taken up the correct position on the vibration platform (trunk erect with hip and knee in slight flexion). In the first 6–8 s, muscle activity was recorded without a WBV stimulus (static standing). Then WBV was applied at the frequency the participant was randomized to, and the EMG was recorded for the next 5 min. The root mean square (RMS) value was determined based on the EMG signal during the period of static standing and WBV, which represented the amplitude of muscle activity. Neuromuscular activation during the exercises was defined as the RMS EMG signal normalized to the peak RMS EMG signal of a MVIC [18], recorded as percentage of EMG_MVIC_.

#### 2.3.3. Frequency Spectrum Analysis of the EMG Signals

The EMG activities of the subjects at 20, 30, and 40 Hz of WBV during standing with slight knee flexion were recorded, and then a spectral analysis of the EMG signal was performed using fast Fourier transformation (FFT). After the frequency spectrum was obtained, a band-stop filter was used to block the interference signal generated by the vibration of the WBV machine. It was also used to block artifacts caused by harmonic vibration at 60 Hz that would coincide with any interference from nearby electrical equipment and power lines. Then, the frequency spectrum of the voluntary muscle contraction during WBV at different frequencies was obtained [19].

### 2.4. Statistical Analysis

All statistical analyses were conducted with SPSS v17.0 software (IBM Corp., Armonk, NY, USA). This study used the chi-square test and one-way ANOVA to analyze whether there were any differences in the basic attributes of the subjects in different age groups. Two-way mixed repeated-measure ANOVA was used to analyze the differences in EMG signals among the subjects of different age groups at 20, 30, and 40 Hz WBV. Then the Scheffe post hoc test was used to compare the differences among various age groups. All tests were two-sided, and the significance level was defined as α = 0.05.

## 3. Results

The basic characteristics of the subjects (including age, sex, and BMI), the RMS of the MVIC, and the static standing of the rectus femoris muscle are shown in Table 1. The subjects included 47 females and 51 males with an average age of 45.1 ± 15.2 years and an average BMI of 23.5 ± 3.3 kg/m^2^, respectively. The average EMG_MVIC_ was 260.6 ± 109.5 μV. The static standing EMGRMS normalized by EMG_MVIC_ was 12.1 ± 7.6%. The data in Table 1 shows statistically significant differences in age, height, and BMI between different groups (*p* < 0.05). Young people had higher EMG_MVIC_ than older people (*p* < 0.001), while SS_EMG_MVIC_ was much higher for older people than for young people (*p* < 0.001).

### 3.1. Comparison between Different Age Groups during Same Frequency WBV

This study found a significant difference in the neuromuscular activation of the rectus femoris muscle (normalized by EMG_MVIC_) among the various age groups at 20 Hz WBV. Muscle contraction was greater in the subjects who were older than in those who were younger (F_(4,93)_ = 82.448, *p* < 0.001). However, at 30 Hz WBV, the difference among the various age groups was not significant (F_(4,93)_ = 2.373, *p* = 0.058). At 40 Hz WBV, there was also a significant difference among the various age groups. Muscle contraction was less in the subjects who were older than in those who were younger (F_(4,93)_ = 18.025, *p* < 0.001) (Table 2).

### 3.2. Interactions of Different Age Groups during WBV at Different Frequencies

Comparisons of the three frequencies (20, 30, and 40 Hz) showed significant differences in the 20–29 years old, 30–39 years old and 40–49 years old age groups: higher frequencies of WBV were related to greater neuromuscular activation (*p* < 0.001), and rectus femoris muscle activity was greater at 30 and 40 Hz than at 20 Hz. In the 50–59 years old age group, there was a significant difference (*p* = 0.006). The between-group comparisons of the three frequencies showed that unlike the muscle activity at 20 and 30 Hz, there was a significant difference at 40 Hz (*p* was 0.018 and 0.005, respectively). The muscle activity was also greater at 20 and 30 Hz than at 40 Hz. In the 60–69 years old age group, there was a significant difference (F = 29.026, *p* < 0.001). The between-group comparisons of the three frequencies showed a significant difference at 20 Hz (*p* < 0.001); the muscle activity at 20 Hz WBV was greater than that at 30 and 40 Hz. However, the between-group comparison of 30 and 40 Hz showed no significant difference (*p* > 0.999), as shown in Figure 1.

### 3.3. Frequency Spectrum Analysis

The spectrum analysis also indicated that there was no significant difference between older subjects and younger subjects during WBV at 20 Hz. However, during WBV at 30 and 40 Hz, the muscle activity patterns of older subjects were different from those of younger subjects. The 50–59 and 60–69 years old age groups had more muscle firing at 20 Hz WBV than 30 Hz or 40 Hz, as shown in Figure 2.

## 4. Discussion

The aim of this study was to investigate the lower limb electromyography activities of different age groups during training with WBV at different frequencies. The results showed that age had a significant effect on rectus femoris EMG activity with WBV at different frequencies. Therefore, it is advised that age be considered in the setting of the frequency parameters of WBV training. Macadam and Feser (2019) recommend a high level of activation (41 to 60% of MVIC) [20], and a very high level of activation (greater than 60% of MVIC). This study found that 20 Hz WBV should be used in older subjects (>50 years old) at a high and very high level of muscle activation, while 40 Hz WBV should be used in younger subjects (<50 years old) at a very high level of activation to strengthen the muscle of the knee extension. Boren et al. (2011) pointed out the exercises that produced greater than 70% of MVIC. These exercises were deemed acceptable for the enhancement of strength [21]. This study found that 20 Hz WBV could induce about 70% of MVIC in the 60–69 years old age group, which might be a good choice as a muscle-strengthening program for older persons. The mixed use of WBV at different frequencies in training programs should be reviewed and possibly changed.

When the WBV training frequency was set at 20 Hz, there was a significant increase in the EMG signals of muscle activations for all age groups as compared with static standing (F = 627.834, *p* < 0.001). This result is consistent with those of other studies [13,22]. Although there is a lack of consistent evidence, it has been generally perceived that during WBV training, the displacement changes generated by the vibration platform cause a sensory reflex and muscle contraction of the Ia muscle spindle. This is considered to be the main cause of motor unit changes [23]. Similar results have been reported in previous studies, reflecting the increase in muscle activity during vibration stimulation [19]. Vibration is an external disturbance to the body’s standing balance. When it is in contact with the body structure, it is sensed by the central nervous system, which in turn regulates the stiffness of the stimulated muscle groups. According to the study of Cardinale and Lim (2003), the muscle activity of the stretch reflex can be considered as a neuromuscular regulatory response that minimizes the effect on soft tissues caused by vibration [6].

The percentage of MVIC values in ages above 50 years old were significantly larger than in age groups below 50 years old at 20 Hz WBV (*p* < 0.001). Previous research results have shown that the balance ability and lower-limb knee extensor muscle strength performance in the elderly are significantly higher after WBV training at 20 Hz than after WBV training at 40 Hz [24]. This study also found that during WBV training at frequencies of 30 Hz and 40 Hz, there was a trend of smaller muscle activity in the older subjects. At the vibration frequency of 40 Hz, the percentage of MVIC values in two groups (the 60–69 age group and the 50–59 age group) were significantly smaller than those in the 40–49, 30–39, and 20–29 age groups (*p* < 0.001). The spectrum analysis figures also showed less EMG activity in the older subjects than in the younger subjects during WBV training at 40 Hz. This result is similar to those of previous studies, suggesting that the training frequency amplitude is unlikely to cause muscle strengthening in healthy young people at a frequency greater than 15 Hz. However, this frequency may be sufficient to cause muscle strengthening in physically weak individuals [11]. Jakobsson (1990) and Verdijk et al. (1990) indicated that the proportion of Type I muscle fibers in the lower-limb muscles is higher in the elderly than in young people, which may be the reason why there is no neuromuscular signal response in the elderly receiving high-frequency WBV training [25,26].

One limitation of this study is that the intensity of WBV (frequency and amplitude) was not investigated. In this study, only the frequency was changed. The amplitude of the WBV was not. However, other studies have verified that there is no dose–response relationship between the intensity of WBV and the optimization of neuromuscular activation and muscle strength [13,27]. In consideration of the research limitations, future studies may lengthen the WBV training time to two or three months, or expand the vibration training to different amplitudes to understand the relative neuromuscular changes in the WBV training of different age groups.

The results of this study are consistent with those of previous studies in that electromyography activity in the elderly did not increase with increases in WBV frequency to 30 Hz and 40 Hz. Previous literature has suggested that studies on surface EMG analysis should be done cautiously during WBV training [19]. The alternating current signal interference (60 Hz) and the vibration frequencies (20, 30, and 40 Hz) of the WBV machine will degrade the quality and reliability of the analysis data [28]. The above signal can be removed using spectrum analysis filtered by a band-stop filter. As proven by Fratini et al. (2009), a relatively correct EMG signal can be obtained after a specific filter is used for proper signal processing [29]. This study also dealt with EMG signals in accordance with this recommendation.

## 5. Conclusions

The research results showed that the muscle activities of the rectus femoris in the 60–69 years old age group at 20 Hz WBV were significantly greater than those in the 20–29, 30–39, 40–49, and 50–59 years old age groups. However, opposite results were obtained at 30 and 40 Hz WBV. As a result, age was found to have a significant effect on the EMG activity during WBV at different frequencies. Therefore, during WBV training, it is necessary to prioritize age when setting the training parameters. It is advised that WBV at 20 Hz be provided in the training of older adults (>60 years old), which may result in more significant neuromuscular responses.

## Figures and Tables

**Figure 1 ijerph-18-11942-f001:**
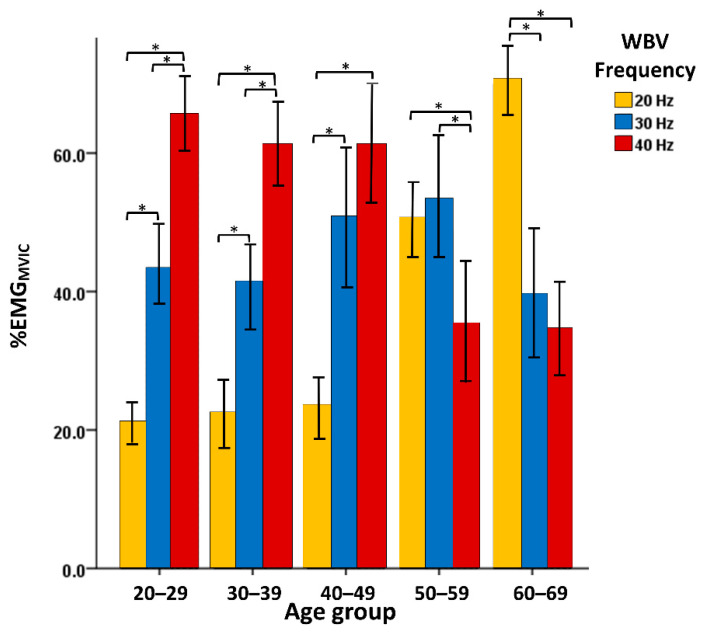
The neuromuscular activation of rectus femoris muscle during 20, 30, and 40 Hz whole-body vibration normalized by EMG_MVIC_. * *p* < 0.05; the values are presented as mean ± 1.96 SE (standard error).

**Figure 2 ijerph-18-11942-f002:**
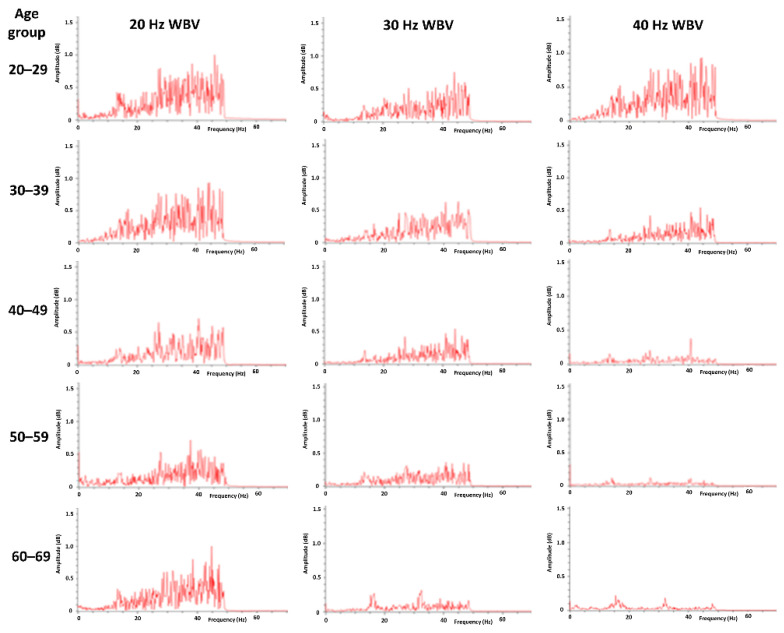
The frequency spectrum of a typical case of rectus femoris muscle EMG signals of different age groups during whole-body vibration at different frequencies.

**Table 1 ijerph-18-11942-t001:** The basic characteristics of the participants (*n* = 98).

	20–29 Years Old Group (*n* = 21)	30–39 Years Old Group (*n* = 18)	40–49 Years Old Group (*n* = 18)	50–59 Years Old Group (*n* = 20)	60–69 Years Old Group (*n* = 21)	*p*-Value
Age (years)	24.2 (22.9–25.5)	34.7 (33.3–36.1)	46.1 (44.9–47.3)	54.0 (52.6–55.4)	65.7 (64.7–66.7)	<0.001 *
Sex (Male/Female)	12/9	9/9	11/7	10/10	9/12	0.811
Height (cm)	168.6 (164.0–173.3)	162.3 (157.5–167.0)	164.1 (160.4–167.9)	164.0 (160.8–167.2)	160.2 (157.2–163.2)	0.024 *
Body Mass (kg)	62.0 (56.4–67.7)	61.2 (54.5–67.9)	68.2 (62.8–73.6)	64.8 (58.9–70.6)	61.9 (57.3–66.5)	0.357
Body Mass Index	21.6 (20.5–22.7)	23.0 (21.4–24.6)	25.3 (23.6–27.0)	23.9 (22.3–25.6)	23.6 (22.7–25.3)	0.007 *
EMG_MVIC_ (μV)	321.4 (289.5–353.4)	296.4 (273.5–369.6)	275.0 (226.0–324.0)	210.8 (167.8–253.8)	182.8 (156.2–209.4)	<0.001 *
SS_EMG_MVIC_ (%)	7.6 (6.0–9.24)	8.5 (6.6–10.4)	8.1 (6.2–9.9)	15.3 (11.8–18.9)	19.0 (15.8–22.1)	<0.001 *

* *p* < 0.05; the values are given as mean (95% confidence interval). EMG_MVIC_: the root mean square EMG of the rectus femoris muscle during maximum voluntary isometric contraction; SS_EMG_MVIC_: the root mean square EMG of the rectus femoris muscle during static standing normalized by the EMG_MVIC_.

**Table 2 ijerph-18-11942-t002:** The neuromuscular activation of the rectus femoris muscle during whole-body vibration training at different frequencies in different age groups.

WBV Frequency	20–29 Years Old Group (*n* = 21)	30–39 Years Old Group (*n* = 18)	40–49 Years Old Group (*n* = 18)	50–59 Years Old Group (*n* = 20)	60–69 Years Old Group (*n* = 21)	*p*-Value
20 Hz	21.4 ^d,e^ (18.0–24.7)	22.2 ^d,e^ (17.1–27.3)	22.7 ^d,e^ (18.3–27.1)	49.7 ^a,b,c,e^ (44.4–54.9)	69.2 ^a,b,c,d^ (62.8–75.7)	<0.001 *
30 Hz	43.7 (38.0–49.4)	40.3 (34.1–46.6)	50.0 (40.4–59.7)	53.3 (44.2–62.3)	39.9 (30.8–49.0)	0.058
40 Hz	64.7 ^d,e^ (58.9–70.6)	60.6 ^d,e^ (54.4–66.8)	60.4 ^d,e^ (51.7–69.1)	35.6 ^a,b,c^ (26.8–44.4)	34.4 ^a,b,c^ (27.2–41.6)	<0.001 *
*p*-Value	<0.001 *	<0.001 *	<0.001 *	0.006 *	<0.001 *	

* *p* < 0.05; the values are given as mean (95% confidence interval); WBV: whole-body vibration. Using the Scheffe test, ^a^: showed a significant difference in the 20–29 years old age group (*p* < 0.001); ^b^: showed a significant difference in the 30–39 years old age group (*p* < 0.001); ^c^: showed a significant difference in the 40–49 years old age group (*p* < 0.001); ^d^: showed a significant difference in the 50–59 years old age group (*p* < 0.001); ^e^: showed a significant difference in the 60–69 years old age group (*p* < 0.001).

## Data Availability

The datasets used and/or analyzed during the current study are available from the study group on reasonable request. Please contact the corresponding author.

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
