# Peer review of "Is 20 Hz Whole-Body Vibration Training Better for Older Individuals than 40 Hz?"

_ijerph, 2021, doi:10.3390/ijerph182211942_

Round 1

Reviewer 1 Report

This study investigates the effects of different WBV frequencies on muscle activation in different age groups.

The study comes to a clear-cut result and recommends age dependent frequencies for WBV.

This reviewer likes to have responses to two aspects:

(1) What about the recruitment of the participants. Any selection bias to report?

(2) Is there a measure to describe any training effect, e.g. thigh circumference or endurance or such.

Minor: Pls, check: Kg should read kg; µv should read µV; throughout? Weight should read body mass (body mass index)

Author Response

Comment 1.  What about the recruitment of the participants. Any selection bias to report?

Authors reply: Thank you for your incisive comments. All subjects were healthy who were hospital volunteers and community members enrolled in the study. None of the subjects had a history of neuromuscular disease, and none of the subjects had a current diagnosis of illness at the time of the investigation. According to their age grouping, divided into 20-29 years old group, 30-39 years old group, 40-49 years old group, 50-59 years old group, 60-69 years old group, a total of 5 groups.  There had no selection bias to report.  (Lines 80-85)

Comment 2.  Is there a measure to describe any training effect, e.g. thigh circumference or endurance or such.

Authors reply: Thank you for your incisive comments. Subjects in this study were randomly given 20, 30, and 40Hz WBV each at once on different day, while recording the electromyography signals of their rectus femoris muscle, so there would be no training effect.

Comment 3. Pls, check: Kg should read kg; µv should read µV; throughout? Weight should read body mass (body mass index)

Authors reply: Thank you for your incisive comments. The correction has been completed in accordance with your comments.

Reviewer 2 Report

This study titled “Is 20 Hz Whole-body Vibration Training Better for Older Individuals than 40 Hz?” was carried out with the aim of “to use WBV at three different frequencies, 20, 30, and 40 21 Hz, with subjects from different age groups to analyze the activation of the rectus femoris muscle”.

The study is interesting and important because the literature still lacks data for the application of vibration training.

With some minor corrections, especially in the results, the work would be ready to be published.

MINOR

ABSTRACT: I suggest replacing the keywords age by age groups; electromyography signals by electromyography; and rectus femoris muscle by rectus femoris; to meet MeSH recommendations (line 32).

RESULTS: The authors could explain where the differences found in the Table 1. Additionally in Table 1, I did not understand what meaning “y/o group” for all age categories. I think that the authors could exclude this of the Table 1, as well as the “y/o” on the x-axis of the Figure 1, in order to make the reading cleaner. Put the “p” of line 172 in lowercase, for standardization. I suggest changing the expression “of different frequencies” to “with different frequencies” in the title of section 3.2. On the other hand, figure 1 does not show what the text says. It is not possible to identify any significant difference in it. The authors need to find a way to show where these differences are (with brackets and asterisks, for example).

DISCUSSION: Insert “(data not shown)” after the final of the first sentence of second paragraph (after “compared with static standing”), since the authors did not show this information in results.

Author Response

Reviewer 2:

Comment 1. ABSTRACT: I suggest replacing the keywords age by age groups; electromyography signals by electromyography; and rectus femoris muscle by rectus femoris; to meet MeSH recommendations (line 32).

Authors reply: Thank you for your incisive comments. The correction has been completed in accordance with your comments.  (Line 32)

Comment 2. RESULTS: The authors could explain where the differences found in the Table 1.

Authors reply: Thank you for your incisive comments. The data in Table 1 shows statistically significant differences in age, height, BMI between different groups (ps < 0.05). Young people had higher EMGMVIC than older people (p < 0.001), while SS_EMGMVIC is much higher for older people than for young people (p < 0.001).   (Lines 145-148)

Comment 3. Additionally in Table 1, I did not understand what meaning “y/o group” for all age categories. I think that the authors could exclude this of the Table 1, as well as the “y/o” on the x-axis of the Figure 1, in order to make the reading cleaner.

Authors reply: Thank you for your incisive comments. "y/o" means "years old" and had been replaced entirely by "years old", and addressed with the word "Age group" in Figure1 and 2.

Comment 4. Put the “p” of line 172 in lowercase, for standardization.

Authors reply: Thank you for your incisive comments. The correction has been completed in accordance with your comments.   (Line 187)

Comment 5. I suggest changing the expression “of different frequencies” to “with different frequencies” in the title of section 3.2.

Authors reply: Thank you for your incisive comments. The correction has been completed in accordance with your comments.   (Line 175)

Comment 6. On the other hand, figure 1 does not show what the text says. It is not possible to identify any significant difference in it. The authors need to find a way to show where these differences are (with brackets and asterisks, for example).

Authors reply: Thank you for your incisive comments. The correction has been completed in accordance with your comments. (in figure 1)

Comment 7. DISCUSSION: Insert “(data not shown)” after the final of the first sentence of second paragraph (after “compared with static standing”), since the authors did not show this information in results.

Authors reply: Thank you for your incisive comments. The correction has been completed in accordance with your comments. ‘When the WBV training frequency was set at 20 Hz, there was a significant increase in the EMG signals of muscle activations for all age groups as compared with static standing (F = 627.834, p < 0.001).‘  (Line 222 )
